# Cardiovascular Risk across Glycemic Categories: Insights from a Nationwide Screening in Mongolia, 2022–2023

**DOI:** 10.3390/jcm13195866

**Published:** 2024-10-01

**Authors:** Nomuuna Batmunkh, Khangai Enkhtugs, Khishignemekh Munkhbat, Narantuya Davaakhuu, Oyunsuren Enebish, Bayarbold Dangaa, Tumurbaatar Luvsansambuu, Munkhsaikhan Togtmol, Batzorig Bayartsogt, Khishigjargal Batsukh, Tumur-Ochir Tsedev-Ochir, Enkhtur Yadamsuren, Altaisaikhan Khasag, Tsolmon Unurjargal, Oyuntugs Byambasukh

**Affiliations:** 1Department of Endocrinology, School of Medicine, Mongolian National University of Medical Sciences, Ulaanbaatar 14210, Mongolia; amd24e007@gt.mnums.edu.mn (N.B.); khishignemekh@mnums.edu.mn (K.M.); altaisaikhan@mnums.edu.mn (A.K.); 2Department of Family Medicine, School of Medicine, Mongolian National University of Medical Sciences, Ulaanbaatar 14210, Mongolia; khangai@mnums.edu.mn; 3State Central Third Hospital, Ulaanbaatar 210648, Mongolia; narantuyad623@gmail.com (N.D.); tsch@shastinhospital.mn (T.-O.T.-O.); 4Ministry of Health, Ulaanbaatar 14253, Mongolia; oyunsuren@moh.gov.mn (O.E.); bayarbold@moh.gov.mn (B.D.); tumurbaatar@moh.gov.mn (T.L.); munkhsaikhan@moh.gov.mn (M.T.); 5Department of Epidemiology and Biostatistics, School of Public Health, Mongolian National University of Medical Sciences, Ulaanbaatar 14210, Mongolia; batzorig@mnums.edu.mn; 6First Central Hospital Mongolia, Ulaanbaatar 210648, Mongolia; khishigjargal@fchm.edu.mn; 7Department of Dermatology, School of Medicine, Mongolian National University of Medical Sciences, Ulaanbaatar 14210, Mongolia; enkhtur@mnums.edu.mn; 8Department of Cardiology, School of Medicine, Mongolian National University of Medical Sciences, Ulaanbaatar 14210, Mongolia

**Keywords:** impaired fasting glucose, cardiovascular disease, central obesity, Mongolia

## Abstract

(1) **Background**: Diabetes mellitus is a significant risk factor for cardiovascular disease (CVD), a leading cause of death globally. Recent studies have highlighted the role of pre-diabetes, particularly impaired fasting glucose (IFG), in elevating CVD risk even before the onset of diabetes. The objective of this study was to assess cardiovascular disease (CVD) risk across specific glycemic categories, including normoglycemia, impaired fasting glucose (IFG), newly diagnosed diabetes, and long-standing diabetes, in a large Mongolian population sample. (2) **Methods**: This cross-sectional study utilized data from a nationwide health screening program in Mongolia between 2022 and 2023, involving 120,266 adults after applying inclusion criteria. The participants were categorized based on fasting plasma glucose levels (NGT): normoglycemia, IFG, newly diagnosed diabetes, and long-standing diabetes. CVD risk was assessed using WHO risk prediction charts, considering factors like age, blood pressure, smoking status, and diabetes status. (3) **Results**: CVD risk varied significantly with glycemic status. Among those with NGT, 62.9% were at low risk, while 31.2% were at moderate risk. In contrast, the IFG participants had 49.5% at low risk and 39.9% at moderate risk. Newly diagnosed diabetes showed 38.1% at low risk and 43.3% at moderate risk, while long-standing diabetes had 33.7% at low risk and 45.9% at moderate risk. Regression analysis indicated that glycemic status was independently associated with moderate to high CVD risk (OR in IFG: 1.13; 95% CI: 1.09–1.18), even after adjusting for age, gender, and central obesity. (4) **Conclusions**: This study emphasizes the need for early cardiovascular risk assessment and intervention, even in pre-diabetic stages like IFG.

## 1. Introduction

Diabetes mellitus is well established as a significant risk factor for cardiovascular disease (CVD), which is the leading cause of morbidity and mortality worldwide [1]. While the role of diabetes in increasing CVD risk is widely acknowledged, pre-diabetes, particularly impaired fasting glucose (IFG), has garnered growing attention in recent years. Pre-diabetes is an intermediate state of hyperglycemia, often preceding the onset of type 2 diabetes mellitus (T2DM); it is characterized by fasting glucose levels that are higher than normal but not yet reaching the threshold for diabetes diagnosis. The progression from pre-diabetes to T2DM is associated with a marked increase in cardiovascular events, even before full-blown diabetes is established [1,2]. The relationship between pre-diabetes and cardiovascular risk is increasingly recognized as crucial to understanding the broader spectrum of glucose metabolism disorders. A total of 129 studies included in a meta-analysis have shown that individuals with IFG already exhibit an elevated risk of CVD compared to normoglycemic individuals, despite not yet having diabetes [3]. This highlights the need to consider pre-diabetes not merely as a precursor to diabetes but as a condition warranting its own intervention strategies due to its independent association with adverse cardiovascular outcomes.

In Mongolia, the prevalence of diabetes has seen a sharp increase from 3.2% in 1999 to 8.3% in 2020, reflecting a concerning public health trend [4,5]. Concurrently, cardiovascular disease has remained the leading cause of death since 1992, accounting for 32.1% of all deaths in 2019 [6]. This rising burden underscores the importance of addressing not only diabetes but also the pre-diabetic states that contribute to the growing cardiovascular disease epidemic in the country. Given Mongolia’s unique population dynamics, including rapid urbanization, significant dietary shifts, and lifestyle changes, the country has experienced rising rates of diabetes and cardiovascular diseases.

While numerous studies have explored the cardiovascular implications of diabetes [1,2,7], fewer have focused on the distinct roles that pre-diabetic conditions, such as IFG, play in driving cardiovascular risk [8,9]. Additionally, the differentiation between old and newly diagnosed diabetes adds another layer of complexity to the understanding of how various stages of glucose dysregulation influence CVD outcomes [10]. This study aims to investigate the relationship between cardiovascular disease (CVD) risk and glycemic status; specifically, it evaluates normoglycemia, impaired fasting glucose (IFG), newly diagnosed diabetes, and long-standing diabetes, using the WHO CVD risk prediction charts in a large Mongolian population.

## 2. Materials and Methods

### 2.1. Data Collection and Study Participants

This cross-sectional study utilized data from a nationwide health screening initiative conducted by the Ministry of Health in Mongolia between 2022 and 2023. Detailed protocols and methodologies for the data collection have been reported elsewhere [11]. In brief, the screening covered 209,055 adults who attended diabetes screening. Participants with missing data related to CVD risk assessment and non-fasting participants were excluded (*n* = 165,730). Further exclusions were made based on data validity, with outliers for fasting blood glucose (FBG), total cholesterol, and waist circumference removed. Outliers were defined as values falling outside the 95% confidence interval or the 2.5th and 97.5th percentiles. This left 120,266 participants’ data available for analysis in this study (Appendix A).

The study was conducted according to the Helsinki Declaration, and it was approved by the medical ethical committee of the Ministry of Health (Approval No: 23/042, dated 5 July 2023).

### 2.2. Glucose Category Evaluation

The participants were asked to fast overnight (for at least 8 h) before having blood samples taken the next morning. Blood glucose measurements were standardized across all screening units using an enzymatic glucose oxidase method in plasma. The participants were classified into glucose categories based on fasting plasma glucose levels, following criteria recommended by the World Health Organization (WHO) [2]. Normal glucose tolerance (NGT) was defined as fasting glucose < 6.1 mmol/L, impaired fasting glucose (IFG) as 6.1–6.9 mmol/L, and diabetes as ≥7.0 mmol/L [1]. Only venous plasma glucose measurements were considered for diagnostic accuracy.

For diabetes categorization, the participants responded to two questions in a tuberculosis risk assessment and a diabetes mellitus (DM) risk assessment. Those who self-reported having diabetes, regardless of glucose levels, were considered as having “existing (old) diabetes”. The participants who had never been diagnosed with diabetes but had glucose levels meeting the criteria for diabetes were classified as having “newly diagnosed diabetes”. The fasting state was confirmed by both the participant responses and the records from diagnostic imaging notes. The participants with IFG (6.1–6.9 mmol/L) were categorized accordingly as IFG.

### 2.3. Cardiovascular Disease (CVD) Risk Assessment

CVD risk was assessed using the WHO CVD risk prediction charts for the Asia region [12]. The assessment incorporated age, gender, systolic blood pressure, smoking status, total cholesterol, and diabetes status. Blood pressure was measured twice with an electronic apparatus at 5 min intervals, and the average was recorded. Total cholesterol levels were measured using an enzymatic colorimetric method with a reference value of <5.18 mmol/L, which was considered normal. CVD risk was categorized into four groups based on the 10-year risk of a cardiovascular event [12]: low risk (<10%), moderate risk (10–19%), high risk (20–29%), and very high risk (≥30%).

Given that the WHO CVD risk charts provide separate models for people with and without diabetes, we applied the appropriate charts based on glycemic status. For the participants in the NGT (normal glucose tolerance) and IFG (impaired fasting glucose) groups, we used the CVD risk chart designed for individuals without diabetes. Conversely, for participants in the newly diagnosed diabetes and pre-existing diabetes (old DM) groups, we used the CVD risk chart specific to individuals with diabetes. This ensured that cardiovascular risk assessment was appropriately tailored to each group based on their glycemic status.

The participants who self-reported a history of stroke or myocardial infarction were automatically classified into the very high-risk category.

### 2.4. Other Variables

Data on the general characteristics were collected during health screenings and included age, gender, education level, marital status, living area (urban vs. rural), smoking status, alcohol use, fruit and vegetable intake, and physical activity. Standard protocols were followed for measurements of body weight, height, waist circumference, and blood pressure. BMI was calculated using height and weight.

For analysis, education level was categorized as “lower” (4 years of education or less) or “above”; marital status was categorized as “married or cohabitant” or “others”; and living area was categorized as urban vs. rural. Smoking status was classified as current (including those who quit within the last 6 months) or never smoked. Alcohol use was defined as consumption within the last 30 days, with thresholds differing for men and women. Fruit and vegetable intake was assessed based on the WHO STEPS criteria, where consuming 5 or more servings per week was considered sufficient [13]. Physical activity was defined as regularly engaging in sports or activities meeting the threshold of 10,000 steps daily [14]. BMI was categorized as normal weight, overweight (25–29.9 kg/m^2^), and obesity (≥30 kg/m^2^) [15]. Central obesity was defined as waist circumference ≥ 90 cm for men and ≥80 cm for women [16].

### 2.5. Statistical Analysis

The characteristics of the study population were expressed as means with standard deviations (SDs) and as percentages with counts according to glucose status categories. Differences between groups were analyzed using ANOVA for continuous variables and Pearson’s chi-square test for categorical variables. Additional analyses were conducted by splitting the participants into categories based on gender, age, and central obesity. The age groups were categorized into tertiles: Group 1 (18–36 years), Group 2 (37–52 years), and Group 3 (53 years and older). Interaction analyses for age, gender, and central obesity were conducted using binary regression analysis. In the regression model, moderate to high CVD risk was used as the dependent variable, with normal glucose tolerance (NGT) serving as the reference group in comparisons with impaired fasting glucose (IFG), newly diagnosed diabetes, and pre-existing diabetes. The results are presented as odds ratios (OR) with corresponding 95% confidence intervals (CI).

Statistical analyses were conducted using IBM SPSS V.28.0, and the graphics were produced using GraphPad Prism 9.0. A *p*-value < 0.05 was considered statistically significant for all tests.

## 3. Results

Table 1 presents the baseline characteristics of the study population, stratified by diabetes categories: normal glucose tolerance (NGT), impaired fasting glycemia (IFG), newly diagnosed diabetes (newly DM), and old diabetes (old DM). The mean age of the total population was 44.3 ± 15.2 years, with a significant increase across the diabetes categories from 42.5 ± 15.3 years in the NGT group to 53.7 ± 12.4 years in the old DM group (*p* < 0.001). Men constituted 39.4% of the overall population, with a higher proportion in the IFG (46.2%) and newly DM (48.0%) groups (*p* < 0.001). Regarding living areas, 41.4% of participants resided in urban areas, but this proportion varied significantly, with only 32.8% of those in the IFG group being urban residents, compared to 59.0% in the old DM group (*p* < 0.001). The educational level was generally high across the population, with 91.4% having middle or higher education levels. However, those in the IFG group had a slightly higher proportion of lower education levels at 10.2% (*p* < 0.001).

BMI increased progressively across the categories, from 26.1 ± 4.7 kg/m^2^ in the NGT group to 29.7 ± 5.5 kg/m^2^ in the newly DM group (*p* < 0.001). The prevalence of obesity followed a similar trend and was highest in the newly DM group at 44.8% (*p* < 0.001). Waist circumference, a key indicator of central obesity, showed significant differences by diabetes category and gender. For males, the mean waist circumference increased from 87.2 ± 13.6 cm in the NGT group to 98.4 ± 14.6 cm in the newly DM group (*p* < 0.001), while for females, it increased from 83.4 ± 13.2 cm to 93.3 ± 14.5 cm (*p* < 0.001). The proportion of individuals with central obesity was highest in the diabetes groups, reaching 78.5% in the newly DM group (*p* < 0.001). Blood pressure levels were also elevated in the diabetes groups, with a mean systolic blood pressure of 128.6 ± 18.4 mmHg in the newly DM group compared to 118.9 ± 15.4 mmHg in the NGT group (*p* < 0.001). Diastolic blood pressure followed a similar pattern (*p* < 0.001). Cholesterol and triglyceride levels were progressively higher across the categories, with triglycerides reaching 2.0 ± 1.4 mmol/L in the newly DM group (*p* < 0.001).

Lifestyle factors, such as smoking and alcohol consumption, also differed by category. Smoking prevalence was 19.5% overall, with a higher prevalence in the IFG (22.4%) and newly DM (23.8%) groups (*p* < 0.001). Alcohol consumption was reported by 9.3% of the total population, with a slightly higher rate of 12.4% in the IFG group (*p* = 0.024). Regular physical activity was reported by 60.0% of the participants, with the highest proportion in the old DM group at 62.9% (*p* < 0.001).

CVD risk can vary based on gender and age. Although gender differences were generally minimal, some slight differences were observed in the middle-aged groups, suggesting that gender alone does not have a major influence on CVD risk.

Figure 1 illustrates the distribution of cardiovascular disease (CVD) risk levels by diabetes category and gender. Among those with normal glucose tolerance (NGT), the majority of the men (62.9%) and women (60.1%) are in the low-risk category. However, as glucose tolerance worsens to IFG, the proportion of individuals in the low-risk category decreases to 49.5% in men and 48.0% in women, while the moderate-risk category rises to 39.9% and 40.8%, respectively. In the newly DM and old DM groups, the low-risk category declines further to 38.1% in men and 36.7% in women. The prevalence of high- (orange) and very high-risk (red) categories increases significantly, particularly in women, where 16.8% fall into the high-risk category compared to 16.5% of men in the newly DM group. In the old DM group, the moderate-risk category is predominant, with 45.9% of men and 46.7% of women falling into this group.

Table 2 displays the distribution of CVD risk levels by age and gender. In Age Group 1 (18–36 years), most of the individuals remain in the low-risk category, which comprises 79.4% of the men and 80.3% of the women. However, as age increases, the proportion of low-risk individuals decreases, with only 43.0% of the men and 43.5% of the women remaining in this category in Age Group 3 (53–94 years). The moderate-risk category increases significantly with age, reaching 43.4% of the men and 44.5% of the women in Age Group 3. High- (orange) and very high-risk (red) categories become more prominent in the older age groups, especially in the men, where 12.4% fall into the high-risk category.

Given that no significant gender differences were observed, but age appeared to play a critical role, we further investigated how age, combined with glucose status, impacts CVD risk. Table 3 explores the distribution of CVD risk levels by diabetes category across the age groups. Among those with normal glucose tolerance (NGT), low-risk status remains predominant in Age Group 1 (79.4%) but decreases to 55.2% in Age Group 3. As glucose tolerance worsens to IFG, the low-risk proportion decreases further, particularly in Age Group 3 (50.1%), while the moderate-risk category rises to 41.0%. In the newly DM and old DM groups, the low-risk category diminishes significantly, with Age Group 3 showing only 35.1% and 25.8% at low risk, respectively. The moderate-risk category becomes dominant in the older age groups, while the high- and very high-risk categories (orange and red) increase significantly, particularly in Age Group 3, where 25.4% of individuals with old DM fall into these categories. These findings suggest that both age and glucose status play pivotal roles in determining CVD risk.

In the regression analysis, the individuals with impaired fasting glucose (IFG) had a significantly higher risk of moderate to high cardiovascular disease (CVD) compared to those with normal glucose tolerance (NGT), with an unadjusted odds ratio (OR) of 1.73 (95% CI: 1.67–1.79). When adjusting for age, the OR decreased to 1.26 (95% CI: 1.22–1.31), indicating that age had a weakening effect on the association. Gender adjustments had minimal impact, with the OR for IFG remaining at 1.24 (95% CI: 1.19–1.28). The presence of central obesity further attenuated the association, reducing the OR for IFG to 1.13 (95% CI: 1.09–1.18). However, despite these adjustments, the association between glycemic status and CVD risk remained statistically significant. Notably, the individuals with pre-existing diabetes had the highest risk, with an unadjusted OR of 3.34 (95% CI: 3.16–3.52) that remained elevated even after full adjustments (OR: 1.94, 95% CI: 1.83–2.05). Even after these adjustments, impaired fasting glucose (IFG) remained an independent predictor of moderate to high CVD risk, with an adjusted odds ratio (OR) of 1.11 (95% CI: 1.06–1.15), confirming the independent contribution of glycemic status to CVD risk (Table 4).

To determine whether central obesity contributes to CVD risk independently of glucose status, we examined the interaction between central obesity, glucose status, and CVD risk. Figure 2 highlights the distribution of the CVD risk levels by diabetes category and the presence of central obesity. Among those with normal glucose tolerance (NGT), 70.4% of the non-central obese group are in the low-risk category, compared to 56.2% of the centrally obese group. As glucose tolerance progresses to IFG, the proportion of low-risk individuals decreases further in the centrally obese group (46.8%). In the newly DM group, only 47% of the non-central obese group remain at low risk, while the proportion drops to 35.7% in the centrally obese group.

The old DM category shows that most of the centrally obese individuals fall into the moderate-risk category (53.2%), with only 32.1% remaining at low risk. These findings indicate that while glucose status plays a significant role in CVD risk, central obesity further exacerbates the risk, particularly in those with impaired glucose tolerance.

## 4. Discussion

The findings of this study strongly suggest that cardiovascular disease (CVD) risk is closely associated with glycemic status, even in the pre-diabetic phase. Our results corroborate existing evidence that individuals with impaired fasting glucose (IFG) already exhibit a significantly higher CVD risk compared to those with normoglycemia, highlighting the critical need for early detection and intervention [2,12]. This reinforces the notion that pre-diabetic conditions, such as IFG, are not merely transitional phases but are associated with substantial cardiovascular risk, necessitating early management to mitigate long-term consequences. This aligns with previous studies that suggest that early intervention in the IFG stage could substantially reduce the progression to diabetes and the subsequent development of cardiovascular complications [17].

The differences in CVD risk observed between newly diagnosed and long-standing diabetes in our study further underscore the progressive and cumulative nature of glucose dysregulation. This is consistent with Seshasai et al.’s findings, which demonstrated that fasting glucose levels exceeding 5.6 mmol/L were significantly associated with increased mortality due to vascular and nonvascular causes, highlighting the broader implications of glucose dysregulation beyond diabetes [18]. The elevated risk in long-standing diabetes, which was notably higher than in newly diagnosed cases, indicates that prolonged exposure to hyperglycemia likely exacerbates vascular damage and accelerates the development of more severe cardiovascular outcomes [10,19,20]. This aligns with the concept that chronic hyperglycemia, through various mechanisms, such as oxidative stress, endothelial dysfunction, and chronic inflammation, contributes to the gradual but steady progression of atherosclerosis and other cardiovascular pathologies. Over time, these mechanisms lead to more severe clinical manifestations, underlining the importance of long-term glucose management in reducing the risk of CVD.

Focusing on a Mongolian population is particularly relevant given the unique epidemiological profile of the region. Rapid urbanization and lifestyle shifts in Mongolia have led to increasing rates of diabetes and related complications, making this an important public health concern [6]. The cultural, dietary, and lifestyle changes in this context are distinct from those of other populations, which may influence the interplay between glycemic status and cardiovascular risk. Our findings indicate that both IFG and diabetes are significant drivers of CVD risk in this population, with central obesity acting as a critical modifier. The data suggest that while both newly diagnosed and long-standing diabetes confer a significant risk of CVD, long-standing diabetes presents an even higher burden, potentially due to the accumulated vascular damage over time [20,21]. This highlights the importance of considering both the duration of diabetes and the presence of central obesity when assessing cardiovascular risk in clinical practice.

Our study demonstrates that cardiovascular disease (CVD) risk is closely tied to glycemic status, with even pre-diabetic individuals showing a significantly higher CVD risk compared to normoglycemic individuals. This aligns with prior studies, such as those by Zuo et al. [17] and Huang et al. [22], which highlight the importance of early intervention in impaired fasting glucose (IFG) stages to mitigate long-term CVD risks [18]. The literature increasingly supports the view that pre-diabetes is not a benign state. Several studies have consistently demonstrated that individuals with IFG are at a substantially higher risk of developing CVD compared to normoglycemic individuals, even after adjusting for traditional risk factors [22,23,24,25]. The underlying mechanisms driving this increased risk likely involve a combination of low-grade systemic inflammation, endothelial dysfunction, and accelerated progression of atherosclerosis in pre-diabetic states [25,26]. These pathophysiological processes are thought to be initiated early in the course of glucose dysregulation and may persist or worsen as individuals transition to overt diabetes. Therefore, early intervention strategies targeting individuals in the pre-diabetic phase could be crucial in preventing the onset of CVD and improving long-term cardiovascular outcomes.

In terms of central obesity, our study found that even among individuals with similar glycemic profiles, those with central obesity exhibited a higher CVD risk. This is in line with existing research indicating that central adiposity, particularly visceral fat accumulation, is a stronger predictor of CVD than general obesity [2,27,28]. The adverse metabolic effects of visceral fat, including increased insulin resistance, dyslipidemia, and pro-inflammatory cytokine production, are well-established contributors to cardiovascular pathology. In pre-diabetic individuals, the combination of central obesity and glycemic dysregulation may lead to a more pronounced risk of CVD, emphasizing the need for comprehensive management strategies that address both metabolic and adiposity-related factors [2,3,9]. Additionally, Zhang et al. [29] emphasized that alternative anthropometric measures like the body roundness index (BRI) further highlight the role of central obesity in predicting all-cause mortality, reinforcing the need for comprehensive management strategies targeting both glycemic dysregulation and visceral fat accumulation [30]. Public health initiatives focusing on weight management, particularly in individuals with central obesity and pre-diabetes, could therefore play a pivotal role in reducing the incidence of CVD.

The distinction between newly diagnosed and long-standing diabetes in this study provides additional insights into the progression of cardiovascular risk. Although both groups are at elevated risk, the individuals with long-standing diabetes exhibited a more pronounced risk of CVD, which was likely due to the cumulative effects of chronic hyperglycemia and associated metabolic derangements. This finding is supported by prior research showing that the duration of diabetes is a key determinant of cardiovascular outcomes, with longer exposure leading to more extensive vascular damage and a greater burden of atherosclerotic disease [20,30]. The concept of ‘metabolic memory’—where early exposure to hyperglycemia has long-lasting effects on cardiovascular health—also supports the need for early and sustained glycemic control to mitigate long-term CVD risk.

Our study draws strength from the large dataset of the Mongolian population, encompassing approximately 2 million adults, with over 100,000 participants included in the final analysis. The extensive sample size enhances the generalizability of our findings within the Mongolian context and provides a robust basis for the evaluation of the association between glycemic status and CVD risk. Nevertheless, several limitations must be acknowledged. The cross-sectional design of the study precludes the establishment of causality between glycemic status and CVD risk. Longitudinal studies are needed to confirm the temporal relationship and causative pathways. Additionally, traditional CVD risk scores, such as the Framingham risk score (FRS), were not applicable due to the unavailability of certain questionnaire components. This limitation arises from the original design of the data collection, which was not specifically intended for CVD risk assessment. A limitation of this study is the use of body muscle index to diagnose overweight and obesity, which may not fully capture the complex relationship between body composition and health outcomes. Recent research, such as the findings on the body roundness index and its association with all-cause mortality [29], highlights the need for more comprehensive anthropometric measures. Another limitation is the diagnosis of diabetes, which was not confirmed using HbA1c or oral glucose tolerance tests (OGTTs). While venous plasma glucose testing is a valid method for diagnosing diabetes, the lack of more comprehensive diagnostic tools could introduce some inaccuracies. To address this, we excluded inappropriate data, such as those of outliers and non-fasting participants, thereby minimizing potential errors. Despite these efforts, a more rigorous diagnostic protocol would have been ideal and could have provided even greater accuracy. Future studies should aim to incorporate more detailed diagnostic measures, longitudinal follow-up, and consideration of other CVD risk factors to provide a more comprehensive understanding of the relationship between glycemic status and cardiovascular risk in this population.

## 5. Conclusions

In conclusion, our findings underscore the need for comprehensive cardiovascular risk assessment in individuals across the glycemic spectrum, including those with IFG and early-stage diabetes. The results suggest that interventions targeting pre-diabetes and central obesity may be crucial in reducing CVD risk in high-risk populations. Given the rising prevalence of diabetes and cardiovascular disease in Mongolia, there is an urgent need to develop targeted public health strategies that address both glycemic control and obesity management to curb the growing burden of cardiovascular morbidity and mortality. 

## Figures and Tables

**Figure 1 jcm-13-05866-f001:**
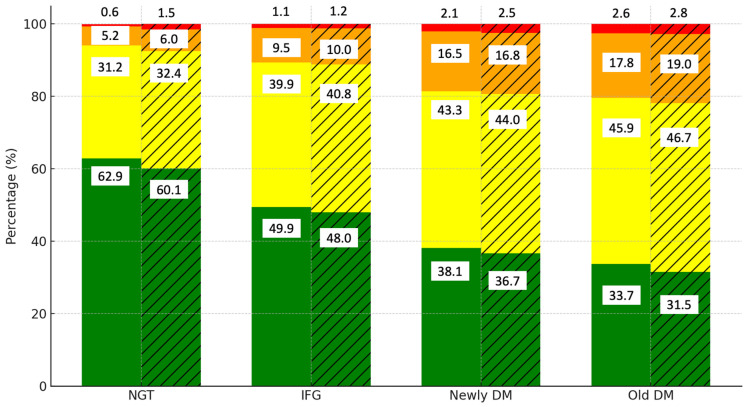
Distribution of CVD risk levels by diabetes category and gender. Notes: green, yellow, orange, and red colors represent low, moderate, high, and very high CVD risk categories, respectively. Solid bars indicate data for men, while hatched (striped) bars represent data for women.

**Figure 2 jcm-13-05866-f002:**
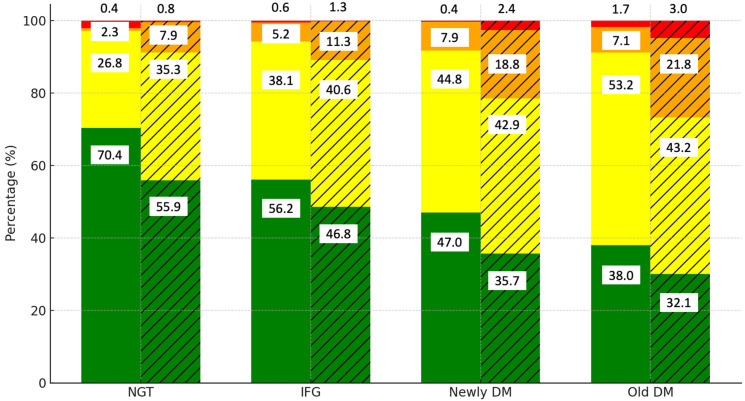
Distribution of CVD risk levels by diabetes category and presence of central obesity. Notes: green, yellow, orange, and red colors represent low, moderate, high, and very high CVD risk categories, respectively. Solid bars indicate data for the non-central obese group, while hatched (striped) bars represent data for the central obese group.

**Table 1 jcm-13-05866-t001:** Characteristics of study population.

Findings	Total	Glycemic Status				
		NGT	IFG	Newly DM	Old DM	*p*-Value
Frequency, n	120,266	71,215	40,080	7990	981	-
Age (years)	44.3 ± 15.2	42.5 ± 15.3	50.5 ± 12.3	52.3 ± 11.8	53.7 ± 12.4	<0.001
Male, % (n)	39.4 (47,370)	37.9 (36,584)	46.2 (7078)	48.0 (1106)	41.9 (2602)	<0.001
Living area: urban, % (n)	41.4 (49,749)	41.7 (40,234)	32.8 (5022)	35.9 (827)	59.0 (3666)	<0.001
Education: lower, % (n)	8.6 (10,351)	8.4 (8126)	10.2 (1562)	7.1 (163)	8.0 (500)	<0.001
BMI (kg/m^2^)	26.7 ± 4.9	26.1 ± 4.7	28.6 ± 4.9	29.7 ± 5.5	29.1 ± 5.3	<0.001
Obesity, % (n)	22.7 (27,318)	19.1 (18,388)	35.8 (5488)	44.8 (1031)	38.8 (2411)	<0.001
Waist Circumference (male, cm)	89.0 ± 14.1	87.2 ± 13.6	94.3 ± 13.9	98.4 ± 14.6	96.8 ± 14.1	<0.001
Waist Circumference (female, cm)	84.9 ± 13.7	83.4 ± 13.2	90.6 ± 13.6	93.3 ± 14.5	92.9 ± 13.8	<0.001
Central Obesity, % (n)	55.9 (67,179)	51.8 (49,979)	70.9 (10,871)	78.5 (1806)	72.8 (4523)	<0.001
Systolic (mmHg)	120.2 ± 16.0	118.9 ± 15.4	124.8 ± 16.9	128.6 ± 18.4	127.0 ± 17.0	<0.001
Diastolic (mmHg)	77.6 ± 10.6	76.7 ± 10.3	80.5 ± 10.8	82.4 ± 11.6	82.0 ± 11.1	<0.001
Total cholesterol (mmol/L)	5.1 ± 1.2	5.0 ± 1.2	5.3 ± 1.3	5.4 ± 1.4	5.5 ± 1.3	<0.001
Triglycerides (mmol/L)	1.3 ± 0.9	1.2 ± 0.8	1.6 ± 1.1	2.0 ± 1.4	1.8 ± 1.2	<0.001
Smoking, % (n)	19.5 (23,494)	18.9 (18,272)	22.4 (3425)	23.8 (549)	20.1 (1248)	<0.001
Alcohol use, % (n)	9.3 (11,168)	8.8 (8478)	12.4 (1900)	10.2 (235)	8.9 (555)	0.024
Fruit and vegetable daily use, % (n)	25.4 (30,499)	25.4 (24,485)	24.9 (3816)	26.1 (600)	25.7 (1598)	0.022
Regular exercise and PA, % (n)	60.0 (72,181)	59.8 (57,690)	60.0 (9201)	59.9 (1380)	62.9 (3910)	<0.001

Data are presented as mean ± SD and percentages (numbers). NGT, normal glucose tolerance; IFG, impaired fasting glycemia; DM, diabetes mellitus. Intergroup *p*-values showed significant differences (*p* < 0.05) for all variables when comparing NGT (normal glucose tolerance) with newly diagnosed diabetes mellitus (newly DM) and long-standing diabetes mellitus (old DM) groups, as well as when comparing IFG (impaired fasting glucose) with newly DM and old DM groups. Additionally, all variables were significantly different (*p* < 0.05) between NGT and IFG.

**Table 2 jcm-13-05866-t002:** Distribution of CVD risk level by age and gender.

Gender	Age Category	N	CVD Risk Category, % (n)				
			Low	Moderate	High	Very High	*p*-Value
Male	T1 (18–36)	18,702	78.2% (14,621)	19.8% (3706)	1.7% (314)	0.3% (61)	<0.0001
	T2 (37–52)	14,085	52.3% (7365)	39.3% (5532)	7.6% (1070)	0.8% (118)	
	T3 (53–94)	14,583	39.0% (5694)	44.5% (6492)	14.8% (2154)	1.7% (243)	
Female	T1 (18–36)	22,393	79.7% (17,848)	18.9% (4225)	1.0% (229)	0.4% (91)	<0.0001
	T2 (37–52)	26,079	59.9% (15,611)	35.0% (9120)	4.6% (1196)	0.6% (152)	
	T3 (53–94)	24,424	41.3% (10,076)	45.1% (11,005)	12.4% (3027)	1.3% (316)	

Data are presented percentages (numbers).

**Table 3 jcm-13-05866-t003:** Distribution of CVD risk level by age, gender, and glucose status.

**Gender and Age**	**Glucose Status**	**N**	**CVD Risk Category, % (n)**				
			**Low**	**Moderate**	**High**	**Very High**	** *p* ** **-Value**
**Male**							
T1 (18–36)	NGT	17,169	79.4% (13,626)	18.9% (3246)	1.4% (246)	0.3% (51)	<0.0001
	IFG	1160	66.6% (772)	29.1% (338)	3.7% (43)	0.6% (7)	
	New DM	118	59.3% (70)	30.5% (36)	10.2% (12)	0.0% (0)	
	Old DM	255	60.0% (153)	33.7% (86)	5.1% (13)	1.2% (3)	
T2 (37–52)	NGT	9811	55.2% (5420)	37.9% (3719)	6.2% (610)	0.6% (62)	<0.0001
	IFG	2959	50.1% (1483)	41.0% (1214)	8.0% (236)	0.9% (26)	
	New DM	433	38.1% (165)	43.6% (189)	16.4% (71)	1.8% (8)	
	Old DM	882	33.7% (297)	46.5% (410)	17.3% (153)	2.5% (22)	
T3 (53–94)	NGT	9604	43.0% (4125)	43.4% (4167)	12.4% (1195)	1.2% (117)	<0.0001
	IFG	2959	35.1% (1038)	47.7% (1411)	15.5% (458)	1.8% (52)	
	New DM	555	27.6% (153)	46.1% (256)	23.2% (129)	3.1% (17)	
	Old DM	1465	25.8% (378)	44.9% (658)	25.4% (372)	3.9% (57)	
	**Glucose Status**	**N**	**CVD Risk Category, % (n)**				
			**Low**	**Moderate**	**High**	**Very High**	** *p* ** **-Value**
**Female**							
T1 (18–36)	NGT	20,799	80.3% (16,695)	18.5% (3838)	0.9% (183)	0.4% (83)	<0.0001
	IFG	1093	76.5% (836)	21.0% (229)	2.2% (24)	0.4% (4)	
	New DM	132	70.5% (93)	22.7% (30)	6.1% (8)	0.8% (1)	
	Old DM	369	60.7% (224)	34.7% (128)	3.8% (14)	0.8% (3)	
T2 (37–52)	NGT	21,161	61.5% (13,012)	34.0% (7191)	4.0% (849)	0.5% (109)	<0.0001
	IFG	3339	57.5% (1920)	36.5% (1219)	5.4% (179)	0.6% (21)	
	New DM	458	46.5% (213)	40.8% (187)	12.0% (55)	0.7% (3)	
	Old DM	1121	41.6% (466)	46.7% (523)	10.1% (113)	1.7% (19)	
T3 (53–94)	NGT	17,884	43.5% (7779)	44.5% (7960)	11.0% (1961)	1.0% (184)	<0.0001
	IFG	3813	40.3% (1538)	44.6% (1699)	13.7% (522)	1.4% (54)	
	New DM	606	30.4% (184)	49.3% (299)	17.2% (104)	3.1% (19)	
	Old DM	2121	27.1% (575)	49.4% (1047)	20.7% (440)	2.8% (59)	

Data are presented as percentages (numbers). NGT, normal glucose tolerance; IFG, impaired fasting glycemia; DM, diabetes mellitus.

**Table 4 jcm-13-05866-t004:** Association of glycemic status with moderate to high CVD risk.

Association of Glycemic Status with Moderate to High CVD Risk	Odds Ratio	95% CI		*p* Value
		Lower Bound	Upper Bound	
Unadjusted				
Normal fasting glycemia	1.0 (reference)	-	-	-
Impaired fasting glycemia	1.73	1.67	1.79	<0.001
Newly diagnosed diabetes	2.75	2.53	2.99	<0.001
Pre-existing diabetes (old)	3.34	3.16	3.52	<0.001
Adjusted for age				
Normal fasting glycemia	1.0 (reference)	-	-	-
Impaired fasting glycemia	1.26	1.22	1.31	<0.001
Newly diagnosed diabetes	1.91	1.75	2.08	<0.001
Pre-existing diabetes (old)	2.19	2.07	2.32	<0.001
Adjusted for age, gender				
Normal fasting glycemia	1.0 (reference)	-	-	-
Impaired fasting glycemia	1.24	1.19	1.28	<0.001
Newly diagnosed diabetes	1.87	1.71	2.04	<0.001
Pre-existing diabetes (old)	2.17	2.05	2.30	<0.001
Adjusted for age, gender, central obesity				
Normal fasting glycemia	1.0 (reference)	-	-	-
Impaired fasting glycemia	1.13	1.09	1.18	<0.001
Newly diagnosed diabetes	1.63	1.49	1.78	<0.001
Pre-existing diabetes (old)	1.94	1.83	2.05	<0.001
Adjusted for age, gender, central obesity, fasting blood glucose level				
Normal fasting glycemia	1.0 (reference)	-	-	-
Impaired fasting glycemia	1.11	1.06	1.15	<0.001
Newly diagnosed diabetes	1.37	1.24	1.52	<0.001
Pre-existing diabetes (old)	1.76	1.65	1.88	<0.001

Regression analysis with glycemic status for CVD risk compared to the reference group (NGT). Data presented as odds ratio with 95% confidence intervals (95% CI). OR, odds ratio.

## Data Availability

The data used to support the findings of this study are available from the corresponding author upon request.

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
