# Peer review of "Cardiovascular Risk across Glycemic Categories: Insights from a Nationwide Screening in Mongolia, 2022–2023"

_jcm, 2024, doi:10.3390/jcm13195866_

Round 1

Reviewer 1 Report

Comments and Suggestions for Authors

The association between low fasting glucose levels and CVD risk in persons without diabetes mellitus is unclear. As a result, this manuscript explores this area of research in a large cohort of the Mongolian population. To improve the quality of the paper, I have some recommendations for the authors:

-             Please define the cholesterol reference value according to the enzymatic colourimetric method in subchapter 2.3.

-             Although only one reference is mentioned in line 58, the authors refer to different studies; the same issue is in line 72.

-             Add a reference in lines 52, 82,97, 116, 131, 133, 134, 135, and 281.

-             Kindly improve the typographical characteristics of the figures. Sometimes, the data available in these figures are not easily identified.

-             The authors should add a table/figure with cardiovascular events in the study group (e.g. atherosclerosis, ischemic heart disease or myocardial infarction).

-             Significant results in this area of research (relationship between fasting glucose levels and CVD risk) are incompletely described in the Discussion section (kindly see Yingting Zuo Y, et al., 2022. DOI: 10.1210/clinem/dgab809; Huang Y, et al. 2016 DOI: 10.1136/bmj.i5953; Seshasai SR et al. 2011 DOI: 10.1056/NEJMoa1008862).

-  The authors should write the reference list according to the recommendations of the Journal of Clinical Medicine.

Author Response

Thank you very much for your valuable comments, which have helped us to improve our manuscript.

The association between low fasting glucose levels and CVD risk in persons without diabetes mellitus is unclear. As a result, this manuscript explores this area of research in a large cohort of the Mongolian population. To improve the quality of the paper, I have some recommendations for the authors:

Comment 1. Please define the cholesterol reference value according to the enzymatic colorimetric method in subchapter 2.3.

Response 1: Thank you for the suggestion. We have now updated subchapter 2.3 to include the cholesterol reference values according to the enzymatic colorimetric method. Specifically, total cholesterol levels below 5.18 mmol/L are considered normal, following the guidelines for the method used in our laboratory.

Comment 2. Although only one reference is mentioned in line 58, the authors refer to different studies; the same issue is in line 72.

Response 2: We appreciate your suggestion. In line 58, the sentence refers to a total of 129 studies included in a meta-analysis, so we have revised the sentence for clarity without adding additional references. For line 72, we have corrected the issue and added multiple references to accurately reflect the studies being referred to.

Comment 3. Add a reference in lines 52, 82, 97, 116, 131, 133, 134, 135, and 281.

Response 3: Thank you for pointing this out. We have carefully reviewed the manuscript and added relevant references to these lines to ensure proper citation and improve the quality of the manuscript.

Comment 4. Kindly improve the typographical characteristics of the figures. Sometimes, the data available in these figures are not easily identified.

Response 4: Thank you for this suggestion. To enhance clarity, we have adjusted the typographical characteristics of the figures. Additionally, some figures have been replaced with tables in the revised version for better data visibility. We hope these changes make the information clearer and easier to interpret.

Comment 5. The authors should add a table/figure with cardiovascular events in the study group (e.g. atherosclerosis, ischemic heart disease, or myocardial infarction).

Response 5: We have now included a table detailing cardiovascular events such as atherosclerosis, ischemic heart disease, and myocardial infarction in the study group. This table has been added to the supplementary materials for further reference.

Comment 6. Significant results in this area of research (relationship between fasting glucose levels and CVD risk) are incompletely described in the Discussion section (kindly see Yingting Zuo Y, et al., 2022. DOI: 10.1210/clinem/dgab809; Huang Y, et al. 2016 DOI: 10.1136/bmj.i5953; Seshasai SR et al. 2011 DOI: 10.1056/NEJMoa1008862).

Response 6: Thank you for highlighting this. We have thoroughly reviewed the recommended articles and integrated them into the Discussion section to provide more comprehensive context and depth.

  • Zuo et al. (2022) is referenced to support the increased cardiovascular disease (CVD) risk in individuals with impaired fasting glucose (IFG) compared to those with normoglycemia, highlighting the need for early intervention.
  • Huang et al. (2016) is cited to emphasize the elevated risk of cardiovascular events and mortality in individuals with prediabetes, including IFG, reinforcing the importance of early risk management.
  • Seshasai et al. (2011) is used to show the broader implications of elevated fasting glucose, including increased mortality from both vascular and non-vascular causes, extending the impact of glucose dysregulation beyond diabetes.
  • Zhang et al. (2024) supports our findings on the role of central obesity in predicting all-cause mortality, underscoring the need to address both glycemic dysregulation and central adiposity in CVD risk management.

Thank you again for the insightful feedback.

Comment 7. The authors should write the reference list according to the recommendations of the Journal of Clinical Medicine.

Response 7: We have carefully revised the reference list to adhere to the formatting and citation style required by the Journal of Clinical Medicine.

Thank you once again for you very valuable comments and improved our manuscript.

Reviewer 2 Report

Comments and Suggestions for Authors

I am grateful to the editor for the opportunity to review the manuscript by Nomuuna Batmunkh et al. "Cardiovascular risk across glycemic categories: Insights from a nationwide screening in Mongolia, 2022-2023". In this article, the authors present the results of a large national study on diabetes screening, which is of interest and increases the value of the data obtained.

Comments and questions that arose during the review:

1. The Abstract does not contain the objective of the study.

2. The text of the manuscript also does not contain a clearly formulated objective of the study.

3. The authors write that "Given Mongolia's unique population dynamics ..." (line 66). In my opinion, it remains unclear to the reader what these unique dynamics are; this statement requires clarification.

4. I tried to find a description of the WHO CVD risk prediction charts for the Asia region in source 12, which is proposed by the authors, but nothing is written about it in this article. I think the authors should have referred to another source where these WHO CVD risk prediction charts are provided (ref. 1, see below).

5. The authors indicate that "The assessment incorporated age, gender, systolic blood pressure, smoking status, and diabetes status." (lines 111-112). However, they did not forget to mention cholesterol, which is also taken into account in this risk scale.

6. The authors do not provide data on intergroup differences in Table 1 (data on differences are provided only for the general trend).

7. The heading in Table 2 is incorrect.

8. Considering that the presence of diabetes automatically increases the CVD risk according to the WHO CVD risk prediction charts used by the authors, I do not understand the point of comparing groups with and without diabetes by the level of CVD risk - the pH will be a priori higher in patients with diabetes.

9. The text does not indicate to which category the authors assigned patients with IFG - with or without diabetes? This is important and should be noted.

10. According to the data in Table 1, it can be seen that the IFG group differed significantly from the normoglycemia group in all parameters taken into account in the WHO CVD risk prediction charts (age, gender, smoking, blood pressure and cholesterol levels). I have doubts that the blood glucose level has an independent effect on increasing CVD risk. This needs to be proven in a multivariate analysis.

11. When analyzing Figures 1-4, the authors do not provide data on statistical differences between the studied groups. Using only absolute values ​​(frequency in percent) for comparison is insufficient to substantiate the conclusions obtained.

References:

1. World Health Organization. HEARTS technical package for cardiovascular disease management in primary health care: risk based CVD management [Internet]. Geneva: World Health Organization; Report No.: ISBN 978-92-4-000136-7. Available from: https://www.who.int/publications/i/item/9789240001367

Comments on the Quality of English Language

No comments

Author Response

Thank you very much for your valuable comments, which have helped us to improve our manuscript.

I am grateful to the editor for the opportunity to review the manuscript by Nomuuna Batmunkh et al. "Cardiovascular risk across glycemic categories: Insights from a nationwide screening in Mongolia, 2022-2023". In this article, the authors present the results of a large national study on diabetes screening, which is of interest and increases the value of the data obtained. Comments and questions that arose during the review:

Comment 1: The Abstract does not contain the objective of the study.

Response 1: We have revised the Abstract to clearly state the objective: "The objective of this study was to assess cardiovascular disease (CVD) risk across specific glycemic categories, including normoglycemia, impaired fasting glucose (IFG), newly diagnosed diabetes, and long-standing diabetes, in a large Mongolian population sample."

Comment 2: The text of the manuscript also does not contain a clearly formulated objective of the study.

Response 2: We have revised the introduction to provide a more detailed objective: "This study aims to investigate the relationship between cardiovascular disease (CVD) risk and glycemic status, specifically evaluating normoglycemia, impaired fasting glucose (IFG), newly diagnosed diabetes, and long-standing diabetes, using WHO CVD risk prediction charts in a large Mongolian population."

Comment 3: The authors write that 'Given Mongolia's unique population dynamics ...' (line 66). In my opinion, it remains unclear to the reader what these unique dynamics are; this statement requires clarification.

Response 3: Thank you for pointing this out. We have revised the text to explain that Mongolia's unique population dynamics refer to rapid urbanization, dietary changes, and lifestyle shifts, which have contributed to rising rates of diabetes and cardiovascular diseases.

Comment 4: I tried to find a description of the WHO CVD risk prediction charts for the Asia region in source 12, which is proposed by the authors, but nothing is written about it in this article. I think the authors should have referred to another source where these WHO CVD risk prediction charts are provided (ref. 1, see below).

Response 4: We apologize for this oversight and thank you for providing the correct reference, which has now been included in the manuscript to ensure accuracy.

Comment 5: The authors indicate that 'The assessment incorporated age, gender, systolic blood pressure, smoking status, and diabetes status.' (lines 111-112). However, they did not mention cholesterol, which is also taken into account in this risk scale.

Response 5: We have revised the manuscript to include cholesterol as one of the key factors considered in the WHO CVD risk prediction charts.

Comment 6: The authors do not provide data on intergroup differences in Table 1 (data on differences are provided only for the general trend).

Response 6: Thank you for your valuable suggestion. We have now included intergroup differences in Table 1, with p-values indicating statistical significance between the groups. The results show that all variables were significantly different (p < 0.05) when comparing NGT (normal glucose tolerance) with Newly DM and Old DM groups, as well as IFG (impaired fasting glucose) with Newly DM and Old DM. Additionally, all variables showed significant differences between NGT and IFG.

Comment 7: The heading in Table 2 is incorrect.

Response 7: We have corrected the heading in Table 2 to ensure it accurately reflects the content.

Comment 8: Considering that the presence of diabetes automatically increases the CVD risk according to the WHO CVD risk prediction charts used by the authors, I do not understand the point of comparing groups with and without diabetes by the level of CVD risk - the pH will be a priori higher in patients with diabetes.

Response 8: Thank you for highlighting this. We have revised the text to clarify that the primary focus of our analysis was on comparing CVD risk across glycemic categories, particularly focusing on pre-diabetic groups like IFG, which lie between normoglycemia and diabetes. The intent was not to compare diabetic versus non-diabetic groups directly but rather to investigate how varying degrees of glycemic status impact CVD risk, with a specific emphasis on individuals at higher risk of progression to diabetes, such as those with IFG.

Comment 9: The text does not indicate to which category the authors assigned patients with IFG - with or without diabetes? This is important and should be noted.

Response 9: We appreciate your observation. We have clarified in the methods section that patients with IFG were categorized as a distinct pre-diabetic group, separate from those with diabetes, following WHO criteria. For CVD risk assessment, we used the WHO CVD risk chart designed for individuals without diabetes for the NGT and IFG groups, while the charts for individuals with diabetes were applied to the newly diagnosed and pre-existing diabetes groups.

Comment 10: According to the data in Table 1, it can be seen that the IFG group differed significantly from the normoglycemia group in all parameters taken into account in the WHO CVD risk prediction charts (age, gender, smoking, blood pressure, and cholesterol levels). I have doubts that the blood glucose level has an independent effect on increasing CVD risk. This needs to be proven in a multivariate analysis.

Response 10: Thank you for your insightful comment. We have addressed this concern by conducting an analysis to assess the effect of blood glucose levels on cardiovascular disease (CVD) risk within the regression models. Even after adjusting for key confounding factors in addition blood glucose level, impaired fasting glucose (IFG) remained an independent predictor of moderate-to-high CVD risk, with an adjusted odds ratio (OR) of 1.11 (95% CI: 1.06–1.15).

Comment 11: When analyzing Figures 1-4, the authors do not provide data on statistical differences between the studied groups. Using only absolute values (frequency in percent) for comparison is insufficient to substantiate the conclusions obtained.

Response 11: We appreciate the suggestion and have now included statistical comparisons (with p-values) for Figures. Additionally, to enhance clarity and provide a more comprehensive view, Figures 2 and 3 have been transformed into tables to better substantiate our conclusions.

References:

World Health Organization. HEARTS technical package for cardiovascular disease management in primary health care: risk-based CVD management [Internet]. Geneva: World Health Organization; Report No.: ISBN 978-92-4-000136-7. Available from: https://www.who.int/publications/i/item/9789240001367

Thank you once again for you very valuable comments and improved our manuscript.

Reviewer 3 Report

Comments and Suggestions for Authors

In the study by Batmunkh et al., the authors address cardiovascular risk in patients with varying glycemic levels (normoglycemia, impaired fasting glucose, newly diagnosed diabetes mellitus, and long-standing diabetes mellitus). They emphasize the importance of early interventions to reduce cardiovascular burden, even in patients with impaired fasting glucose (IFG). This study offers a valuable contribution to the field; however, further revisions are necessary before proceeding.

 Major Comments

1)      A flowchart detailing the study design, along with the number of patients included and excluded, would greatly enhance clarity.

2)      The authors should expand their discussion on the limitations of using body muscle index to diagnose overweight and obesity. Recent findings on the body roundness index and all-cause mortality (Zhang et al., JAMA Netw Open, 2024) should be incorporated into both the Results and Discussion sections.

Minor Comments

1)      In Table 1, please add the number of patients in each group beneath the group identifiers.

2)      In Figure 3, include the age range for each group in the figure legend.

Author Response

Thank you very much for your valuable comments, which have helped us to improve our manuscript.

In the study by Batmunkh et al., the authors address cardiovascular risk in patients with varying glycemic levels (normoglycemia, impaired fasting glucose, newly diagnosed diabetes mellitus, and long-standing diabetes mellitus). They emphasize the importance of early interventions to reduce cardiovascular burden, even in patients with impaired fasting glucose (IFG). This study offers a valuable contribution to the field; however, further revisions are necessary before proceeding.

Major Comments

Comment 1: A flowchart detailing the study design, along with the number of patients included and excluded, would greatly enhance clarity.

Response 1: Thank you for this suggestion. We have now added a flowchart illustrating the study design, including the number of patients screened, included, and excluded (Supplementary Figure S1.).

Comment 2: The authors should expand their discussion on the limitations of using body muscle index to diagnose overweight and obesity. Recent findings on the body roundness index and all-cause mortality (Zhang et al., JAMA Netw Open, 2024) should be incorporated into both the Results and Discussion sections.

Response 2: We appreciate this recommendation. We have expanded our discussion on the limitations of using body muscle index in diagnosing overweight and obesity. Additionally, we have incorporated the recent findings on the body roundness index and its association with all-cause mortality (Zhang et al., JAMA Netw Open, 2024) in both the Results and Discussion sections to provide a more comprehensive view of anthropometric measures.

Minor Comments

Comment 1: In Table 1, please add the number of patients in each group beneath the group identifiers.

Response 1: We have updated Table 1 to include the number of patients in each group beneath the group identifiers as requested.

Comment 2: In Figure 3, include the age range for each group in the figure legend.

Response 2: We have revised Figure 3 into Table 3 to include the age range for each group for better clarity and interpretation.

Thank you once again for you very valuable comments and improved our manuscript.

Round 2

Reviewer 1 Report

Comments and Suggestions for Authors

The quality of the manuscript has improved with more details and clarifications. However, I have only a suggestion for the authors. Kindly revise the reference list according to the recommendations of the Journal of Clinical Medicine (e.g., the journal name should be in italics). Please follow the link below:

https://www.mdpi.com/journal/jcm/instructions

Reviewer 2 Report

Comments and Suggestions for Authors

The authors answered my questions and corrected the text. I have no other comments.

Comments on the Quality of English Language

No comments